# RPVC: A Revocable Publicly Verifiable Computation Solution for Edge Computing

**DOI:** 10.3390/s22114012

**Published:** 2022-05-25

**Authors:** Zi Jiao, Fucai Zhou, Qiang Wang, Jintong Sun

**Affiliations:** Software College, Northeastern University, Shenyang 110169, China; 1910470@stu.neu.edu.cn (Z.J.); wangq3635@126.com (Q.W.); 2010510@stu.neu.edu.cn (J.S.)

**Keywords:** publicly verifiable computation, revocable group signature, outsource computing, edge computing

## Abstract

With publicly verifiable computation (PVC) development, users with limited resources prefer to outsource computing tasks to cloud servers. However, existing PVC schemes are mainly proposed for cloud computing scenarios, which brings bandwidth consumption or network delay of IoT devices in edge computing. In addition, dishonest edge servers may reduce resource utilization by returning unreliable results. Therefore, we propose a revocable publicly verifiable computation(RPVC) scheme for edge computing. On the one hand, RPVC ensures that users can verify the correct results at a small cost. On the other hand, it can revoke the computing abilities of dishonest edge servers. First, polynomial commitments are employed to reduce proofs’ length and generation speed. Then, we improve revocable group signature by knowledge signatures and subset covering theory. This makes it possible to revoke dishonest edge servers. Finally, theoretical analysis proves that RPVC has correctness and security, and experiments evaluate the efficiency of RPVC.

## 1. Introduction

The rapid popularization of smart devices has spawned a large number of Internet of Things (IoT) applications, one of which is the Internet of Vehicles (IoV). The reason why vehicles tend to outsource computing tasks that include road conditions and vehicle information to cloud servers during their travel is that the computing resources are limited. Edge computing can improve the response speed and user experience. As a bridge between users and cloud servers, on the one hand, they improve response speed by sharing part of the cloud computing, while users with the limited resource can rely on them to reduce computing pressure. One specific implementation is the Intelligent Transport System (ITS) [1,2], as shown in Figure 1, which is used to help users receive the best driving plan in current road and traffic conditions as soon as possible. There are four participants in ITS: User, road side unit (RSU, which can be seen as an edge server), cloud server, and car manufacturers. The car manufacturer dispatches functions for making a driving plan for the cloud server. The edge server downloads the function from the cloud server. The user sends vehicle parameters to the edge server. The edge server returns results to the user.

However, ITS has the following problems: (1) The cloud server may tamper with the functions uploaded by the car manufacturer, and the edge server may provide users with incorrect results  [3]. (2) When a user is driving, the vehicle needs to switch among RSUs that serve different areas. To verify signature messages from a specific RSU, a large public key list is needed  [4]. This results in overhead storage for users and overhead computation for finding public keys (3) Once a user receives messages from an edge server that it has never met, frequent communication brought by public key transmission will cause delays (4) If the identity of the edge server is exposed, adversaries can use the same attack method to threaten edge servers with similar configurations.

From the example of IoV, the requirements for edge computing are as follows: (1) Results returned by the edge server should be verifiable, and a dishonest edge server can be revoked. (2) The time for the user to verify the result and the number of keys stored should be minimized. (3) Key transmission processes between users and new edge servers should be minimized. (4) The identity of the edge server should be anonymous to users.

For requirement 1, verifiable computation (VC) [5] can be used to ensure the result is correct. However, the verifier in VC schemes can only be the user or the one he specified. Other participants cannot believe in the verification processes or the reliability of results. Therefore, Parno [6] first proposed publicly verifiable computation (PVC) to solve these defects. Since then, Fiore [7] expanded PVC to evaluate the higher-degree polynomial and matrix multiplication. Catalano [8] introduced a one-way function and RSA mathematical hypothesis to improve the computing speed. However, the verification process of the former uses low-efficiency bilinear pairing, and the practical implementation of the latter is very complex. Polynomial commitment [9] achieves two basic goals: making a commitment to a polynomial and providing proof that a specific point belongs to the polynomial. Therefore, the polynomial commitment can be used to improve the efficiency of existing PVC solutions. To revoke dishonest edge servers, James [10] applies the revocable key policy attribute encryption [11,12] to PVC. However, such schemes are based on time-consuming operations such as encryption and decryption, as meanwhile, the revoking process will cause other participants to synchronize the key list. In addition, the latest research  [13,14,15] requires either a trusted computing environment such as SGX or specific hardware support, thus, the scope of their application is limited. Therefore, revocable group signatures are recommended to revoke dishonest edge servers.

For requirements 2–4, group signature schemes are suitable. That is because any group member can make signatures stand for the whole group, and anyone outside the group cannot forge the signature. Verifiers can verify the signature with only one group public key. The verifier only knows that the signature is from a member of the group, but cannot distinguish the specific signer. The group manager can open a group signature to trace the specific signer. When applied to an edge computing scenario, edge servers can form a group and set up a group manager. For users, only one group public key is required to verify any edge server signed results, thus, reducing delay and key storage. There will be no key transmission process between users and the new edge server, moreover, the identity of the edge server is anonymous to users. The group manager can trace the signature of incorrect results to find the dishonest edge servers, so, a revocable group signature is recommended for revoking their computing ability.

The classical group signature scheme proposed by Camenisch [16] cannot revoke group members. To make group signature revocable, Song [17] proposed a revocable group signature scheme to ensure forward security. However, the time cost increases linearly with the number of group members. Camenisch [18] proposed an accumulator solution, but once the group members join or quit the group frequently, the members still in the group need to update their credentials continually. Inspired by Boneh [19], Brickell [20] presents a revocation list (RL) solution that keeps members in the group from frequently updating their credentials. However, the final signature of this scheme contains nine parts, which leads to the extremely tedious verification process. Moreover, the drawback of the latest research  [21] is that there is not an extremely strong privacy demand in an IoV scenario, which will cause resource waste. At the same time, ref. [22,23] based on merkle hash tree, suggested that the storage and computational overhead vary superlinearly along with the number of users who frequently join or quit An attribute tree using secret sharing [24] and Lagrange interpolation impels the users satisfying certain attributes and can decrypt messages under the broadcast encryption [25]. The idea of subset covering or subset difference [26,27] in an attribute tree to reduce search time and communication cost can be used to improve revocable group signature.

Our contributions are as follows:We propose a revocable publicly verifiable computation (RPVC) model. Its main ideas are: Using the properties of PVC to ensure the results returned by the edge server are reliable. Using the properties of group signature to reduce the cost of verification and key storage for users, and keep edge server identity anonymous. If the group signature is revocable, the group manager can trace and revoke the dishonest edge server.After analyzing the RPVC threat model, four security goals of the RPVC model are summarized according to possible attack methods and available information for adversaries: function binding, result reliability, anonymity, and revocability.An RPVC scheme is given. The scheme speeds up the PVC proof generation and verification time with the help of polynomial commitment and improves the revocable group signature with a subset covering idea. Finally, the correctness analysis and security proof of the scheme are provided.We implemented the RPVC scheme, and experiments show that the time delay and storage cost of the RPVC scheme is acceptable when it is applied to edge computing scenarios.

## 2. Related Works

### 2.1. Publicly Verifiable Computation (PVC)

Verifiable computing (VC) was proposed to verify the outsource computing results by Gennaro [28] via a boolean circuit in 2010. Benabbas [29] expands VC to compute polynomials in a higher degree. Other studies such as  [30,31,32] also consider VC However, the common defect of all the above VC schemes is that the verifier can only be the user or the one he specifies, which limits the promotion of VC. Thus, public verifiable computation by Parno [6] was first proposed to address this shortcoming. Though  [33,34] can also achieve PVC, ref. [33] needs an honest user to generate the main private key, which makes the status of users unequal. In addition, the users in  [34] obtain a decrypt key by interaction, which leads to low efficiency. Fiore [7], based on the solution from Benabbas, constructed a PVC scheme aimed to solve matric products. However, the verification time is long because of the use of bilinear mapping. Although Catalano [8] introduced one-way hash function and RSA assumption to improve the speed, this is hard to deploy on existing applications. Another solution to achieve efficient PVC is the work by Ding [15], his idea is to use a trusted computation environment provided by Intel SGX. Similarly, a scheme  [13,14] by Fraust and Liu also needs specific hardware support, these schemes are not suitable for complex large-scale networks.

### 2.2. Revocable Group Signature

Any group member can make group signature stands for the whole group, others cannot forge a group signature. The verifier can verify a group signature by a group public key without finding out the specific signer. The group manager can open the group signature and figure out who makes this signature. The original concept was proposed by Camenisch [16], but this scheme cannot revoke group members. In order to make a group signature revocable, Song [17] developed with a revocable group signature model which can guarantee forward security, but its verification time increases linearly with the number of group members. Camenisch also proposed a scheme based on an accumulator [18], but if group members frequently join or quit the group, another group member needs to update their credential in a timely manner. The same problems also occur in the scheme  [22,23] by Yehia and Buser, furthermore, the mechanism of merkle tree makes the scale increase super-linearly with the joining or quitting of users. Brickell [20] put forward a solution based on the local revocation list, which is inspired by the work of Boneh [19]; group members do not need to update their credentials frequently. However, the final signature contains nine parts, and the verification processes are extremely complex. To confirm whether the signer is in the revocation list more efficiently, Nakanishi [35] brings in a subgroup idea, however, his scheme is still based on an accumulator with the same defects as the Camenisch solution. Yue [21] proposed a revocable group signature which can preserve the privacy, but the drawback is the high consumption of computation resources due to the high level security assumption.

## 3. Preliminaries

### 3.1. Bilinear Maps and Related Assumptions

Let G be cyclic additive group, whose order is prime *p*. G is generated by *g*. Define $G_{T}$ as multiplicative group with the same order *p*. The bilinear pairing $e:G\times G⟶G_{T}$ holds three properties: bilinearity, non-degeneracy, and computability. ε denotes negligible value.

DL Assumption: Given *g* and $a\overset{\$}{\leftarrow }Z_{p}^{*}$, for every adversary A${}_{DL}$, Pr[A${}_{DL}$$(g,g^{a})=a]=\varepsilon$.t-polyDH Assumption [26]: Let $\alpha \overset{\$}{\leftarrow }Z_{p}^{*}$, given a (t+1)−tuple $\langle g,g^{\alpha },g^{\alpha^{2}},\cdots ,g^{\alpha^{t}}\rangle \in G^{t+1}$ as input, for every adversary A${}_{t-polyDH}$,Pr[A${}_{t-polyDH}$$\left(g,g^{\alpha },g^{\alpha^{2}},\cdots ,g^{\alpha^{t}}\right)=\langle \phi \left(x\right),g^{\phi (x)}\rangle ]=\varepsilon$, where $\phi \left(x\right)\in Z_{p}\left[x\right]$.t-SDH Assumption [36]: Let $\alpha \overset{\$}{\leftarrow }Z_{p}^{*}$, given a (t+1)−tuple $\langle g,g^{\alpha },g^{\alpha^{2}},\cdots ,g^{\alpha^{t}}\rangle \in G^{t+1}$ as input, for every adversary A${}_{t-SDH}$,Pr[A${}_{t-SDH}$$\left(g,g^{\alpha },g^{\alpha^{2}},\cdots ,g^{\alpha^{t}}\right)=\langle c,g^{\frac{1}{\alpha +c}}\rangle ]=\varepsilon$, for any value of $c\in Z_{p}\backslash \left\{-\alpha \right\}$.t-BSDH Assumption [26]: Let $\alpha \overset{\$}{\leftarrow }Z_{p}^{*}$, given a (t+1)−tuple $\langle g,g^{\alpha },g^{\alpha^{2}},\cdots ,g^{\alpha^{t}}\rangle \in G^{t+1}$ as input, for every adversary A${}_{t-BSDH}$,Pr[A${}_{t-BSDH}$$\left(g,g^{\alpha },g^{\alpha^{2}},\cdots ,g^{\alpha^{t}}\right)=\langle c,e{\left(g,g\right)}^{\frac{1}{\alpha +c}}\rangle ]=\varepsilon$, for any value of $c\in Z_{p}\backslash \left\{-\alpha \right\}$.

### 3.2. Signature of Knowledge

The signer can use the signature of knowledge (SKSIG) to prove he owns a secret without leaking that secret. It is a kind of non-interactive zero-knowledge prove system, which has three typical constructions: (1) signature of knowledge of discrete logarithms (SKLOG). (2) signature of knowledge of double discrete logarithms (SKLOGLOG). (3) signature of knowledge of an e−th root of the discrete logarithms (SKROOTLOG). No adversary can recover the secret or create an illegal signature by the chosen message attack. More detail is in [16,37].

SKLOG: SKLOG of element $y\in G_{n}$ to the base *g* on message *m* is a pair $\left(c,s\right)\in {\left\{0,1\right\}}^{k}\times Z_{n}^{*}$ satisfying $c=H(m||y||g||g^{s}y^{c})$. H(·) is a one-way hash function. SKLOG is denoted $SKLOG\left[\alpha :y=g^{\alpha }\right]\left(m\right)$, where α is the target of zero-knowledge proof, it should be secret to verifier. For any adversary A${}_{SKLOG}$, Pr[A${}_{SKLOG}$$\left(c,s\right)=\{\alpha \vee \left(c^{\prime },s^{\prime }\right)\}]=\varepsilon$, where $\left(c,s\right)\neq \left(c^{\prime },s^{\prime }\right)$.SKLOGLOG: SKLOGLOG is denoted $SKLOGLOG\left[\beta :y=g^{\alpha^{\beta }}\right]\left(m\right)$, where β should be kept secret to the verifier, other references are public.SKROOTLOG: SKROOTLOG is denoted $SKROOTLOG\left[\beta :y=g^{\beta^{e}}\right]\left(m\right)$, where β should be kept secret to the verifier, other references are public.

### 3.3. Strong RSA Assumption

Let p,q be two big prime integers, compute n=pq. Given tuple (n,e), for every adversary A${}_{RSA}$, Pr[A${}_{RSA}$(n,e)=(z,d)]=ε such that $z^{d}=e\;\mathrm{mod}\;n$ [38].

## 4. Revocable Publicly Verifiable Computation (RPVC) Model

This section first introduces the RPVC model, then provides its threat model, and finally puts forward the design goals.

### 4.1. RPVC Model

As shown in Figure 2, there are four entities in the RPVC model: cloud server, edge server, auditor, and user. The reason why the function owner does not become an RPVC entity is that the edge server downloaded the computing function from a cloud server. The process that the function owner entrusts the computing function to the cloud can be initialized offline Edge servers and the auditor are in the same group, the edge server has the role of a group member, and the auditor has the role of the group manager. RPVC entities are described as follows:Cloud Server: The cloud server receives functions initialized by different function owners and allows legal edge servers to download functions.Edge Server: The edge server sends a request to the auditor for joining edge computing. After the auditor approves, the edge server downloads functions from the cloud server and performs computing for users.Auditor: The auditor is responsible for approving the edge server’s join request and revoking a dishonest edge server who provided incorrect results.User: The user verifies the results returned by the edge server. If the result fails to pass the verification, the user will send a revoke request to the auditor.

As shown in Figure 2, an RPVC can be divided into three phases: an initialize phase, a join phase and an outsource computing phase:Initialize Phase: As step ➀ in Figure 2, this phase can be performed offline by the function owner. The function owner selects the function private key α, and sends the computing function *F* and function evaluation key EK to the cloud server. Then, the function owner sends the verification key VK to users.Join Phase: The join phase includes step ➁–➄ in Figure 2. The edge server applies to the auditor for joining computation in step ➁. After the auditor validates the edge server’s application, the auditor sends a group member certificate Cert or its mask $\sigma_{Cert}$ to the edge server by a secure channel in step ➂. In step ➃ the edge server downloads the computing function *F* and evaluation key EK from the cloud server. In step ➄, the auditor broadcasts data structure *T* which stores the currently valid edge server and group public key Gpk to users.Outsource Computing Phase: The outsource computing phase includes step ➅–➈ in Figure 2. In step ➅, the user sends function input *x* to the edge server. In step ➆, the edge server evaluates function with input *x*. Then, return result y=F(x), corresponding proof proof and computation signature $SKSIG_{comp}$ to the user. In step ➇, the user verifies $SKSIG_{comp}$ to confirm the result is returned by a legal edge server, next, verify *y* is correct with proof. In step ➈, If the result verification doesn’t pass, the user sends a revoke request to the auditor. The auditor traces and revokes a dishonest edge server and reorganize *T* which gets rid of the information of the dishonest edge server.

A revocable publicly verifiable computation scheme RPVC consist of five algorithms (Setup, Register, Compute, Verify, Revoke) as follows:Setup$\left(1^{\lambda },\alpha ,F\right)⟶\left(EK,VK,Gsk,Gpk,T,L\right)$: The algorithm is used for model setup, which includes cloud server setup and auditor setup.
(1)Cloud server setup: In order to make the outsourcing computation results verifiable and so they cannot be forged, the function owner privately selects a random number α, and generates a function evaluation key EK and a function verification key VK according to the security parameter λ.(2)Auditor setup: The auditor generates a group private key Gsk and a group public key Gpk by security parameter λ, that Gsk used for issuing a group member certificate and Gpk used for verifying the validity of group signature for results and its proof. The auditor creates a binary tree *T* which can quickly search all valid edge servers, and record the identities of edge servers in list *L* privately.Register$\left(\sigma_{id},mem\left(\sigma_{id}\right),SKSIG_{Reg},T\right)⟶\left(\sigma_{Cert},T\right)$: The auditor executes this algorithm. The auditor will receive a request from an edge server that wants to join the outsource computation. That request should use the mask of the identity private key $\sigma_{id}$ to prevent the real identity id of the edge server from being exposed. The edge server should use Gpk to generate $mem(\sigma_{id})$ which is the mask code of group membership for the auditor to trace the signer. To prevent man-in-the-middle attacks, the edge server should use knowledge signature $SKSIG_{Reg}$ to prove he knows the identity private key id without leaking it. After the auditor verified $SKSIG_{Reg}$, the edge server will receive the mask of the group member certificate $\sigma_{Cert}$. Then, the auditor adds the edge server to *T* and *L*. The edge server gets the group member certificate Cert by decoding $\sigma_{Cert}$.Compute$\left(x,F,EK,SK_{S},Cert\right)⟶\left(y,proof,SKSIG_{comp}\right)$: The edge server executes this algorithm. The edge server evaluates the function value *y* by the user input *x*, then, uses EK to compute the corresponding proof proof of *y*. Finally, the edge server makes a revocable group signature $SKSIG_{comp}$ to *y* and proof, with Cert and a signature private key set $SK_{S}$. $SKSIG_{comp}$ can prove the identity of the edge server through non-interactive zero-knowledge proof, at the same time, it can ensure the edge server is not revoked by the auditor.Verify$\left(T,Gpk,VK,y,proof,SKSIG_{Comp}\right)⟶\tau_{y}$: The user executes this algorithm. The user first verifies $SKSIG_{comp}$ by *T* and Gpk to ensure the result is from a legal edge server that has not been revoked. Next, the user verifies *y* is correct by VK and proof. Finally, if these two verifications are passed, the user outputs accept token $\tau_{y}=true$, otherwise, $\tau_{y}=false$.Revoke$(T,L,\tau_{y},SKSIG_{Comp})⟶T$: The auditor executes this algorithm. The auditor opens $SKSIG_{comp}$ with *L* to trace the identity of the dishonest edge server under the condition of $\tau_{y}=false$, then removes it from *T* and *L*. From then on, the result returned by a dishonest edge server will never pass the verification.

### 4.2. Threat Model

For users, the auditor is trusted and other entities are semi-trusted. In other words, edge servers and cloud servers may tamper with or forge content. Based on the information available to adversaries, we consider the following two threat models:(1)Chosen Plaintext Attack Model: In this model, the attacker may obtain encryptions of his chosen messages, such as the mask code of id, VK of the function *F*, or the proof for computing result *y*.(2)Chosen Message Attack Model: In this model, the attacker may obtain signatures of his chosen messages, such as additional information which would be used to construct an existential universal forgery group signature.

### 4.3. Design Goals

To achieve RPVC in edge computing, we aim to achieve the following design goals.

(1)Function Binding: The VK and EK should only be used to verify or compute the specific function *F* which the function owner provided. The function binding experiment ${EXP}_{RPVC}^{fb}\left(A\right)$ is shown in Figure 3, the RPVC is *Function Binding* if ${Adv}_{RPVC}^{fb}\left(A\right)$ is negligible for any adversary A.(2)Result Reliability: For the user’s specific input *x*, the edge server should not give valid results and proofs other than the real function value *y*. The result reliability experiment ${EXP}_{RPVC}^{rr}\left(A\right)$ is shown in Figure 3, the RPVC is *Result Reliability* if ${Adv}_{RPVC}^{rr}\left(A\right)$ is negligible for any adversary A.(3)Anonymity: The user should not recover the identity id of any edge server. The anonymity experiment ${EXP}_{RPVC}^{ano}\left(A\right)$ is shown in Figure 4, the RPVC is *Anonymity* if ${Adv}_{RPVC}^{ano}\left(A\right)$ is negligible for any adversary A.(4)Revocability: The user should not accept the results which are returned by revoked edge servers. In addition, the adversary should not show the valid signature associated with wrong results or proofs The revocability experiment ${EXP}_{RPVC}^{rev}\left(A\right)$ is shown in Figure 4, the RPVC is *Revocability* if ${Adv}_{RPVC}^{rev}\left(A\right)$ is negligible for any adversary A.

## 5. Proposed RPVC Scheme

We now give the detailed construction of each algorithm in RPVC. Notations used in RPVC are in Table 1.

### 5.1. Setup

The setup algorithm of the RPVC scheme including cloud server setup and auditor setup.

(1)Cloud server setup: The function owner owns a polynomial form function F=Φ(x)∈Zp[x] with degree deg(Φ)≤t. Φ(x) which can be expressed as Equation (Equation 1)
(1)Φ(x)=∑j=0deg(Φ)ϕjxj
Step 1.The function owner chooses two groups G and GT with prime order *p*, two groups can make bilinear maping e:G×G⟶GT satisfies the t−SDH assumption. G=<e,G,GT> is defined as a bilinear group with generator g2←RG.Step 2.The function owner privately chooses α←RZp*, then, computes function evaluation key EK={G,g2,g2α,⋯,g2αt} and function verification key in Equation (Equation 2) according to α.
(2)VK=∏j=0deg(Φ)(g2αj)ϕjStep 3.The function owner sends *F* and EK to the cloud server, sends VK to users. This step can be completed offline, such as VK can be embedded in vehicle OBU by car manufacturers in IoV applications.(2)Auditor setup: The auditor outputs group private key Gsk and group public tree Gpk with security parameters a,λ, then generates a subset covering complete tree (SCST) according to valid edge servers at time *t*, at last, updates list *L*.
Step 1.The auditor first privately chooses two big primes at random, and gets their product nc as RSA modulus. Then, it generates group G1 in order nc with generator g1←G1. Next, it selects security parameters *a* and λ for knowledge signature. After that, it choses ec←RZn and computes dc which satisfies Equation (Equation 3). Finally, the auditor keeps a group private key Gsk=(nc,dc), and broadcasts a group public key Gpk={nc,ec,g1.G1,a,λ}.
(3)ecdc≡1modφ(nc)Step 2.Let *N* be the overall set of edge servers, *R* is the set of revoked edge servers, clearly, N\R is the set of valid edge servers right now. The auditor builds a minimum complete binary tree CT with a height of l=⌈log|N|⌉, at the same time, it initializes all leaf nodes to ⊥. The root node of CT is recorded as x0,0, other nodes can be expressed as xi,j, where i∈[0,⋯,l], j∈[1,⋯,2i]. According to the subset covering theory, the parent node can be used to represent the set composed of its two child nodes under the condition of both child nodes belonging to N\R. If iterate to the root node in this way, N\R can be expressed by the set of parent nodes, these parent nodes represent num disjoint subsets S1∪S2∪⋯∪Snum, in which num is the minimum amount of disjoint subsets in the current valid leaf node arrangement. The resulting SCST is the set of nodes from the above processes. Algorithm 1 shows how SCST is generated.Step 3.The auditor should assign random ei,j to each node xi,j on SCST, and calculate di,j which satisfies Equation (Equation 4) and then attach a timestamp *t* to SCST. At last, it should put the edge servers’ information into *L*.
(4)ei,jdi,j≡1modφ(nc)

**Algorithm 1** SCST Generator.**Input:** All signers set *N*, revoked set *R*
 1:Build complete binary tree CT with *N* 2:tmp=<> 3:**for all***x* in CT **do** 4:    **if** *x* in *R* **then** 5:        tmp.add(path(x)) 6:    **end if** 7:**end for** 8:**for all***x* in tmp **do** 9:    **if** $x_{left}$ in tmp
**then** add $x_{left}$ to SCST10:    **else** add $x_{right}$ to SCST11:    **end if**12:**end for**13:**return**SCST


### 5.2. Register

The auditor issues a group membership certificate to the edge server and broadcasts the latest SCST to users.

Step 1.The edge server privately selects an identifier $id\overset{R}{\leftarrow }Z_{p}^{*}$, then, computes Equation (Equation 5) as the mask of id and Equation (Equation 6) as the mask of group membership. Next, it makes the knowledge signature $SKSIG_{Reg}$ to $\sigma_{id}$ and $mem(\sigma_{id})$ by Equation (Equation 7), and sends a request for joining the edge computing to the auditor. The request should involve $\sigma_{id}$, $mem(\sigma_{id})$ and $SKSIG_{Reg}$.
(5)$$\sigma_{id}=a^{id}\;mod\;n_{c}$$
(6)$$mem\left(\sigma_{id}\right)=g_{1}^{\sigma_{id}}$$
(7)$$\left\{\begin{array}{c} m=(\sigma_{id}\|mem\left(\sigma_{id}\right))\\ SKSIG_{Reg}=SKLOGLOG\left[id:mem\left(\sigma_{id}\right)=g_{1}^{\sigma_{id}}\right]\left(m\right). \end{array}\right.$$Step 2.If the auditor verifies $SKSIG_{Reg}$ successfully, it indicates that the edge server owns id. Based on this premise, auditor selects free leaf node $x_{l,k}$($k\leq 2^{i}$) on SCST, chooses random $e_{l,k}$ and $d_{l,k}$ which satisfies Equation (Equation 8). Obviously, the group member certificate for the edge server is Equation (Equation 9). The auditor puts the information of the edge server into *L*, the form of the record is $\left\{\sigma_{id},mem\left(\sigma_{id}\right),d_{l,k}\right\}$, after that, it updates SCST by Algorithm 1.
(8)$$e_{l,k}d_{l,k}\equiv 1\;mod\;\varphi \left(n_{c}\right)$$
(9)$$\sigma_{Cert}={\left(\sigma_{id}+e_{l,k}\right)}^{d_{c}}\;mod\;n_{c}$$Step 3.The auditor broadcasts the latest SCST to users, transmits $\sigma_{Cert}$ and $Cred_{id}=g_{1}^{d_{l,k}}$ which is used for proving the identity of the edge server is valid at time *t* to the edge server.Step 4.The edge server creates a signature private key set $SK_{S}=\left\{id,\sigma_{Cert},Cred_{id}\right\}$.

### 5.3. Compute

The edge server evaluates the function value and its proof for users, then, makes the group signature revocable on result and proof.

Step 1.The user sends function input *x* to the edge server.Step 2.The edge server evaluates the function value y=F(x), and calculates the corresponding proof by EK with Equation (Equation 10).
(10)$$proof=g_{2}^{\psi_{x}\left(\alpha \right)},\;\;\psi_{x}\left(\alpha \right)=\frac{F(\alpha )-F(x)}{\alpha -x}\in Z_{p}\left[x\right]$$Step 3.In order to show the validity and that it has not been revoked, the edge server should make the signature *q* to *y* and proof by Equation (Equation 11), where h(·) is a one-way hash function.
(11)$$q={\left(g_{1}^{d_{l,k}}\right)}^{h(y||proof)}\;mod\;n_{c}$$Furthermore, the edge server makes a group signature by Equation (Equation 12) as in [16,37]. The final computation signature is $SKSIG_{Comp}=\left\{q,\tilde{g},\tilde{z},V_{1},V_{2}\right\}$.
(12)$$\left\{\begin{array}{c} \tilde{g}=g_{1}^{r},\;r\overset{R}{\leftarrow }Z_{p}^{*},\;\tilde{z}={\tilde{z}}^{\sigma_{id}}\\ V_{1}=SKLOGLOG\left[id:\tilde{z}={\tilde{g}}^{a^{id}}\right]\left(y\|proof\right)\\ V_{2}=SKROOTLOG\left[Cert:\tilde{z}{\tilde{g}}^{e_{l,k}}={\tilde{g}}^{Cert^{e_{c}}}\right] \end{array}\right.$$Step 4.The edge server returns $\left\{e_{l,k},y,proof,SKSIG_{Comp}\right\}$ to user.

### 5.4. Verify

The user first verifies whether the result is from a valid legal edge server, then, verifies that the result is correct. If a fault occurs during the verification process, the user will send a revoke request to the auditor.

Step 1.The user synchronizes the SCST from the auditor and confirms $e_{l,k}\in SCST$ by Equation (Equation 13).
(13)$$g_{1}^{h(y||proof)}=q^{e_{l,k}}\;mod\;n_{c}$$Step 2.The user rapid verifies $V_{1},V_{2}$ in $SKSIG_{Comp}$ by the hash functions provided by SKLOGLOG and SKROOTLOG.Step 3.The user verifies the result is correct by Equation (Equation 14), and outputs an accept token $\tau_{y}=accept$ after the result hass passed verification. Otherwise, the user sends a revoke request to the auditor; the request contains a reject token $\tau_{y}=reject$ and $SKSIG_{Comp}$.
(14)$$e\left(VK,g_{2}\right)=e\left(proof,\;g_{2}^{\alpha }/g_{2}^{x}\right)\;\cdot \;{e(g_{2},g_{2})}^{F(x)}$$

### 5.5. Revoke

The auditor opens the revocable group signature to lock the identity of the dishonest edge server, removes it from *L*, and broadcasts the updated SCST which deletes the leaf node of the dishonest edge server to users.

Step 1.After the auditor received the revoke request, the auditor opened $SKSIG_{Comp}$ to trace the dishonest edge server with the help of *L*, and then deletes it from *L*.Step 2.The auditor updates SCST by means of deleting the leaf node corresponding to the dishonest edge server via Algorithm 1, then broadcasts the latest SCST. For example, as shown in Figure 5, there are eight signers $x_{3,1},\cdots ,x_{3,8}$. When the auditor receives the request to revoke $x_{3,2},x_{3,5},x_{3,6}$ at time *t*, SCST will be updated to $\left\{t||x_{3,1},x_{2,2},x_{2,4}\right\}$.

## 6. Scheme Analysis

This section first illustrates the correctness of RPVC from the correctness of the result and signature. Then, combined with the security model proved the security of RPVC from the aspects of function binding, result reliability, anonymity, and revocability.

### 6.1. Correctness

#### 6.1.1. Correctness of Result

The edge computing results and its verification processes are correct, because:(15)$$\begin{array}{cc} \; & e\left(proof,g_{2}^{\alpha }/g_{2}^{x}\right)\;\cdot \;{e\left(g_{2},g_{2}\right)}^{F\left(x\right)}\\ \; & =e\left(g_{2}^{\psi_{x}\left(\alpha \right)},g_{2}^{\alpha -x}\right)\;\cdot \;e{\left(g_{2},g_{2}\right)}^{F(x)}\\ \; & ={e\left(g_{2},g_{2}\right)}^{\psi_{x}\left(\alpha \right)\left(\alpha -x\right)+F\left(x\right)}\\ \; & ={e\left(g_{2},g_{2}\right)}^{\frac{F\left(\alpha \right)-F\left(x\right)}{\left(\alpha -x\right)}\left(\alpha -x\right)+F\left(x\right)}\\ \; & ={e\left(g_{2},g_{2}\right)}^{F\left(\alpha \right)}=e\left(VK,g_{2}\right) \end{array}$$

#### 6.1.2. Correctness of Signature

The signature *q* used by the edge server to prove it has not been revoked and the verification process of *q* is correct, because:(16)$$\begin{array}{cc} \; & q^{e_{l,k}}\;mod\;n_{c}\\ \; & =g_{1}^{d_{l,k}h\left(y||proof\right)e_{l,k}}\;\;mod\;n_{c}\\ \; & =g_{1}^{h(y||proof)} \end{array}$$

### 6.2. Security Analysis

The proof method of RPVC uses “game-playing” technology which was proposed in [39,40,41]. This technology uses the game sequence specification to prove that the possibility of the adversary winning the game is negligible, and the probability of the adversary winning two adjacent games should be controlled within a negligible range (i.e., indistinguishable in polynomial time). Define the probability of **Game**
***i*** happens is $Pr(S_{i})$.

#### 6.2.1. Proof of Function Binding

**Game** **0:**This is the original function binding game in Figure 3, A trying to find another $F^{\prime }$ which has the same VK with *F*. A obtains EK, VK and *F* as information. Clearly, ${Adv}_{RPVC}^{fb}\left(A\right)=Pr\left(S_{0}\right)$.

**Game** **1:**This game is the same as Game 0 except that replace $F^{\prime }$ with $F+\hat{F}$. If A can find $F^{\prime }$, he must find a different function $\hat{F}$. So, $Pr\left(S_{1}\right)=Pr\left(S_{0}\right)$.

**Game** **2:**This game is the same as Game 1 except that replace the winning condition to $\hat{\Phi }\left(\alpha \right)=0\;⋀\;{\hat{\phi }}_{j}$ not all 0. $\hat{\Phi }\left(\alpha \right)=0$ since
$$\hat{VK}=g_{2}^{\hat{\Phi }\left(\alpha \right)}=g_{2}^{\Phi^{\prime }\left(\alpha \right)-\Phi \left(\alpha \right)}=g_{2}^{\Phi^{\prime }\left(\alpha \right)}/g_{2}^{\Phi (\alpha )}=1.$$Because $F^{\prime }\neq F$, the polynomial coefficients ${\hat{\phi }}_{j}$ cannot be all 0. Clearly, $Pr\left(S_{2}\right)=Pr\left(S_{1}\right)$.

**Glaim** **1:**$$Pr\left(S_{2}\right)\leq {Adv}_{g,g^{a}}^{DL}\left(B^{A}\right)$$
Let *n* be the degree of polynomial, algorithm B is a tool that assumes A can solve a class of problems including the games. If the simplest case in these problems is the current difficult mathematical problem, it means that the adversary cannot break the security characteristic. $B^{A}\left(n,F,F^{\prime }\right)$ computing a collision as in the following steps:(1)$\alpha^{\prime }\leftarrow A\left(n,\Phi \left(\alpha \right),\Phi^{\prime }\left(\alpha \right)\right)$, 2≤n≤t(2)If $\alpha =\alpha^{\prime }$ return 1Else return 0

**Proof.** *Game 2* is equivalent to $B^{A}\left(t,F,F^{\prime }\right)$. It will be the simplest polynomial of degree one problem when n=2, so ${Adv}_{n=t}^{fb}\left(B^{A}\right)\leq {Adv}_{n=2}^{fb}\left(B^{A}\right)$. That means to solve the latter is easier than the former, A will own more advantages. Now let n=2, the processes to find a collision are as follows:
$$g^{\left(\phi_{1}-\phi_{1}^{\prime }\right)\alpha^{\prime }+\phi_{0}-\phi_{0}^{\prime }}=1$$
$$\Leftrightarrow \left(\phi_{1}-\phi_{1}^{\prime }\right)\alpha^{\prime }+\phi_{0}-\phi_{0}^{\prime }=0$$
$$\Leftrightarrow \alpha^{\prime }=\frac{\phi_{0}^{\prime }-\phi_{0}}{\left(\phi_{1}-\phi_{1}^{\prime }\right)}$$Clearly, find $\alpha^{\prime }$ via $g_{2}$ and $g_{2}^{\alpha }$ on cyclic group G is a more efficient way, which exactly is DL assumption. Hence,
$${Adv}_{RPVC}^{fb}\left(A\right)\leq {Adv}_{n=2}^{fb}\left(B^{A}\right)\leq {Adv}_{g_{2},g_{2}^{\alpha }}^{DL}\left(B^{A}\right)\leq \varepsilon$$That is, the probability of the adversary successfully attacking is negligible, the RPVC scheme achieves *Function Binding*. □

#### 6.2.2. Proof of Result Reliability

**Game** **0:**This is an original result reliability game in Figure 3, A trying to find malicious $y^{*}$ and $proof^{*}$ that can pass the user verification. Clearly, ${Adv}_{RPVC}^{rr}\left(A\right)=Pr\left(S_{0}\right)$.

**Game** **1:**The adversary can compute $\frac{1}{\alpha -x}$. The reason why $Game\;0\approx_{p}Game\;1$ and $Pr\left(S_{1}\right)=Pr\left(S_{0}\right)$ is:$$Verify\left(VK,\;y^{*},{proof}^{*}\right)=Verify\left(VK,\;y,proof\right)$$
(17)$$\begin{array}{cc} \Leftrightarrow {e(g_{2},g_{2})}^{\psi_{x}\left(\alpha \right)\left(\alpha -x\right)}\;\cdot \;{e(g_{2},g_{2})}^{y} & ={e(g_{2},g_{2})}^{\psi_{x}^{*}\left(\alpha \right)\left(\alpha -x\right)}\;\cdot \;{e(g_{2},g_{2})}^{y^{*}}\\ \Leftrightarrow \psi_{x}\left(\alpha \right)\left(\alpha -x\right)+y & =\psi_{x}^{*}\left(\alpha \right)\left(\alpha -x\right)+y^{*}\\ \Leftrightarrow \frac{\psi_{x}\left(\alpha \right)-\psi_{x}^{*}\left(\alpha \right)}{y^{*}-y} & =\frac{1}{\alpha -x} \end{array}$$

**Game** **2:**For $g_{2}$ is part of Gpk, the adversary can give valid pair $\left(-x,g_{2}^{\frac{1}{\alpha -x}}\right)$. Obviously, $Game\;1\approx_{p}Game\;2$.

**Glaim** **2:**$$Pr\left(S_{2}\right)\leq {Adv}_{EK}^{t-SDH}\left(B_{2}^{A}\right)$$$B_{2}^{A}\left(EK,V\right)$ can give a valid pair $\left(c,g_{2}^{\frac{1}{\alpha +c}}\right)$, where *V* is some kind of valid algorithm, $c\in Z_{n}^{*}$. $B_{2}^{A}$ is a bridge between Game2 and t-SDH difficult mathematical.

**Proof.** When $EK=\{G,g_{2},g_{2}^{\alpha },\cdots ,g_{2}^{\alpha^{t}}\}$, algotithm $B_{2}^{A}$ is t-SDH assumption. Though the adversary can give a valid algorithm $V(x,y,proof,y^{*},{proof}^{*})$, ${Adv}_{EK,V}^{t-SDH}\left(B_{2}^{A}\right)\leq {Adv}_{EK}^{t-SDH}\left(B_{2}^{A}\right)$. The output of algorithm *V* is $\left(-x,g_{2}^{\frac{1}{\alpha -x}}\right)$, where
$$g_{2}^{\frac{1}{\alpha -x}}={\left(\frac{proof}{{proof}^{*}}\right)}^{\frac{1}{y^{*}-y.}}$$So the following equation holds,
$${Adv}_{RPVC}^{rr}\left(A\right)\leq {Adv}_{EK}^{t-SDH}\left(B_{2}^{A}\right)\leq \varepsilon$$That is, the probability of the adversary succesfully attacking is negligible, the RPVC scheme achieves *Result Reliability*. □

#### 6.2.3. Proof of Anonymity

**Game** **0:**This is an original anonymity game in Figure 4, A trying to figure out the identity of the edge server. Clearly, ${Adv}_{RPVC}^{ano}\left(A\right)=Pr\left(S_{0}\right)$.

**Game** **1:**The adversary has the ability to extract id from at least one of the following parts: $\tilde{z},V_{1},V_{2}$. Explicitly, Game0⇔Game1 and $Pr\left(S_{1}\right)=Pr\left(S_{0}\right)$ for the identity information in $SKSIG_{Comp}$ only including $\tilde{z},V_{1},V_{2}$.

**Game** **2:**This game is the same as Game 1 except that the adversary has the ability to extract id from $V_{1}$ or $V_{2}$. Function $F_{1}\left(y,x\right)$ denotes the probability of extract *x* from $y=g^{x}$ under the DL assumption. Extract id from $\tilde{z}$ should sequentially execute: $F_{1}\left(\tilde{z},r\sigma_{id}\right)$, $F_{1}\left(\tilde{g},r\right)$ and $F_{1}\left(\sigma_{id},id\right)$. The recursive proof method can refer to the literature [41]. The above process can be expressed as:$$|Pr\left(S_{1}\right)-Pr\left(S_{2}\right)|\leq Pr\left(extract\;id\;from\;\tilde{z}\right),$$
and it is not more than
$$Pr\left(F_{1}\left(\tilde{z},r\sigma_{id}\right)\right)Pr\left(F_{1}\left(\tilde{g},r\right)\right)Pr\left(F_{1}\left(\sigma_{id},id\right)\right)\leq {\left({Adv}_{g,g^{a}}^{DL}\left(\;\cdot \;\right)\right)}^{3}$$Finally, $Game\;1\approx_{p}Game\;2$ and $Pr\left(S_{2}\right)=Pr\left(S_{1}\right)$, due to
$${\left({Adv}_{g,g^{a}}^{DL}\left(\;\cdot \;\right)\right)}^{3}\leq \varepsilon^{3}\leq \varepsilon$$

**Game** **3:**This game is the same as Game 2 except that the adversary can extract id from $V_{1}$. Function $F_{2}\left(y,x\right)$ and $F_{3}\left(y,x\right)$ denotes the probability of extract *x* from *y* under the RSA assumption and SKROOTLOG signature. Extract id from $V_{2}$ should sequentially execute: $F_{3}\left(V_{2},\sigma_{cert}\right)$, $F_{2}\left(\sigma_{cert},\sigma_{id}\right)$ and $F_{1}\left(\sigma_{id},id\right)$. Similiar to Game 2, it can infer that
$$|Pr\left(S_{3}\right)-Pr\left(S_{2}\right)|\leq Pr\left({Event}_{3}\right)\leq {Adv}_{g,g^{a}}^{DL}\left(\;\cdot \;\right){Adv}_{n,e}^{RSA}\left(\;\cdot \;\right){Adv}_{SKSIG}^{SKROOTLOG}\left(\;\cdot \;\right)\leq \varepsilon ,$$
so $Game\;3\approx_{p}Game\;2$ and $Pr\left(S_{2}\right)=Pr\left(S_{3}\right)$. The RPVC scheme achieves *Anonymity* for
$$Pr\left(S_{3}\right)\leq {Adv}_{SKSIG}^{SKLOGLOG}\left(\;\cdot \;\right)\leq \varepsilon$$

#### 6.2.4. Proof of Revocability

**Game** **0:**This is an original revocable game in Figure 4, a revoked A trying to succesfully sign results or faking an honest user’s signature on wrong results. Clearly, ${Adv}_{RPVC}^{rev}\left(A\right)=Pr\left(S_{0}\right)$.

**Game** **1:**This game is the same as Game 0, the adversary can recovery corresponding $d_{l,k}$ by $e_{l,k}$ or replace valid *q* in malicious $SKSIG_{Comp}$. Game0⇔Game1.

**Game** **2:**This game is the same as Game 1 except that the adversary can recover corresponding $d_{l,k}$ by $e_{l,k}$. If an adversary can replace valid *q* in malicious $SKSIG_{Comp}$, he must make sure $V_{2}$ can be verified succesfully So
$$|Pr\left(S_{1}\right)-Pr\left(S_{2}\right)|\leq {Adv}_{SKSIG}^{SKROOTLOG}\left(\;\cdot \;\right)\leq \varepsilon ,$$
further, $Game\;1\approx_{p}Game\;2$. The RPVC scheme achieves *Revocability* for
$$Pr\left(S_{2}\right)\leq {Adv}_{n,e}^{RSA}\left(\;\cdot \;\right)\leq \varepsilon$$

## 7. Performance Analysis

In Table 2, we compare other existing group signature schemes in the IoV scenario with RPVC. The results in Table 2 show that our scheme uses a superior audit method to find the dishonest participants, and the core cryptographic algorithm is the non-interactive zero-knowledge signature, which is mainly based on a hash function that is more efficient than existing schemes. Besides, RPVC updates the SCST at a regular time, which provides participants with more fault tolerance.

In order to compare the performance of the RPVC more intuitively, we conduct a series of experiments to evaluate the cost and efficiency of the RPVC. The experimental environment is deployed on a PC with Ubuntu 20.0.4 TLS, bilinear pairing rely on bn256 (github.com/ethereum/go-ethereum/crypto/bn256/cloudflare, accessed on 20 March 2022), other libraries including PBC 0.5.14 (https://crypto.stanford.edu/pbc/, accessed on 15 March 2022) and GMP-6.2.1 (https://gmplib.org/, accessed on 15 March 2022).

Some basic assumptions in the experiments are as follows: The service radius of RSU is about 2.5 km [47], users’ vehicle speed is not more than 180 km/h. 3G network speed is about 300 KB/s, 4G network speed is about 2.4 MB/s [48]. The reaction time of a driving human to brake is 600–1400 ms [49].

The test contains two parts: One is the process of the edge server applying to join edge computing and the auditor revokes a dishonest edge server, the other one is the user asking for outsourcing computing and receiving verifiable reliable results. The former has three test items: (1) The execution time for the auditor. (2) The size of SCST which the user received from the auditor. (3) The storage space consumed by the user. The latter also has three test items: (4) The extra cost for the edge server to apply RPVC. (5) The time consumed for user verification. (6) The total time delay after applying RPVC.

For test item 1, the execution time for the auditor can be divided into the time to add the edge server into the group and the time to generate SCST. As shown in Table 3, the time to add an edge server into the group is about 27.996 ms, which is independent of the scale of the edge server. As shown in Table 4, with edge servers scale increase in the group, the time of the auditor adding or removing an edge server increases proportionally. However, even if the number of edge server reaches $2^{15}$ (RSU service can cover about 321,700 km${}^{2}$), the auditor can generate SCST within 1 ms. The driving distance is only 1.45 m during the user vehicle receives SCST at the highest speed, far less than the service radius of RSU. That is, users have enough time to safely synchronize the current valid edge server. For test item 2, as shown in Table 4, the size of SCST is independent from the scale of the edge server, SCST is only about 5 kB. For test item 3, the local storage space for the user is multiplied with the increase of the edge server scale, which is shown in Table 5. However, even when the number of edge servers comes up to $2^{15}$, storage is less than 15 MB.

In test items 4–6, we set the degree of polynomial and input *x* as independent variables, the time cost as dependent variable (default is ns, $10^{-9}$s). The rule to choose the independent variable *x* is: randomly select a value from each range, ranges including $\left[0,2^{4}\right],\left[2^{4},2^{5}\right],\left[2^{5},2^{6}\right],\left[2^{6},2^{7}\right],\left[2^{7},2^{8}\right]$. Results of test item 4 are shown in Figure 6a, the extra cost of applying the RPVC proportionally tothe polynomial degree, the larger the *x*, the smaller the curve fluctuation. For test item 5, as shown in Figure 6b, the time of user verification fluctuates between 36 ms and 38 ms, which is less affected by independent variables. Figure 6c indicates the total extra time delay brought by the RPVC application. Even if the degree of a polynomial function is up to 100, the total delay is less than 100 ms, which is far less than the driver’s reaction time [49].

From the above six test items, it is clear that the RPVC can be used to improve the security of existing edge computing applications. We can summarize the key influencing factor from Figure 7: if the polynomial degree is larger than 40, the performance of the edge server takes the most portion of total time delay, the portion gets larger with the increase of degree. So, a better edge server may expand the application scope of the RPVC.

## 8. Discussion

For the requirements of edge computing in the IoV scenario, the RPVC first achieves the goal of results returned by the edge server being verifiable. At the same time, the identity of the edge server is anonymous to user vehicles and a dishonest edge server can be revoked. From the test results, when a new edge server takes part in outsourced computing, user vehicles do not need to exchange keys with it. The time in which the auditor adds one edge server into the group can be fixed, nearly 28 ms that is independent of the scale of the edge server. The time of user vehicles receiving SCST mainly depends on the communication delay because the generated speed of SCST is less than 1 ms. Though the total delay for user vehicles increases with the degree of the polynomial, it is less than 95 ms when the degree is up to 100 (a very complex computation). Furthermore, the storage overhead is acceptable for user vehicles, even if the number of edge servers comes up to $2^{15}$, storage demand is less than 15 MB.

The low delay and overhead are owed to the subset covering complete tree and non-interactive zero-knowledge signature. SCST makes user queries faster than iterating local revoke lists at a small cost. Besides, the non-interactive zero-knowledge signature is mainly based on the hash function, which is more efficient than other large number or exponent multiply schemes. The practical applications of the research can be used to assist the construction of intelligent transportation or vehicle networking.

For future work, we will first reduce the size of SCST for the larger scale of the edge server. Next, machine learning and federated learning can be introduced to improve the performance of edge servers, good solutions can be found in [50,51,52]. In addition, different regions have different traffic rules and habits, these should be considered. Finally, we will extend the outsource function to varied forms, such as verifiable matrix computation.

## 9. Conclusions

In this article, we proposed an RPVC model for the edge computing scenario which can be used in IoV applications. The RPVC model cannot only ensure the results returned by edge servers are reliable, but can also revoke dishonest edge servers. The following security proofs show that the RPVC has characteristics of function binding, result reliability, anonymity, and revocability. Experiments show that a new edge server which takes part in edge computing does not need transfer keys to users, and an auditor can approve the request in a fixed time (28 ms). Due to the SCST, users have a low overhead storage and a faster query time, even when the number of edge servers came up to $2^{15}$, storage demand is less than 15 MB. Because of the non-interactive zero-knowledge signature, even the degree of outsource function up to 100, the total delay of users is about 95 ms. Thus, applying RPVC to existing IoV applications is acceptable. In the future, we are committed to reducing the size of SCST, trying to introduce machine learning or federated learning to improve the performance of edge servers and supporting verifiable matrix computation.

## Figures and Tables

**Figure 1 sensors-22-04012-f001:**
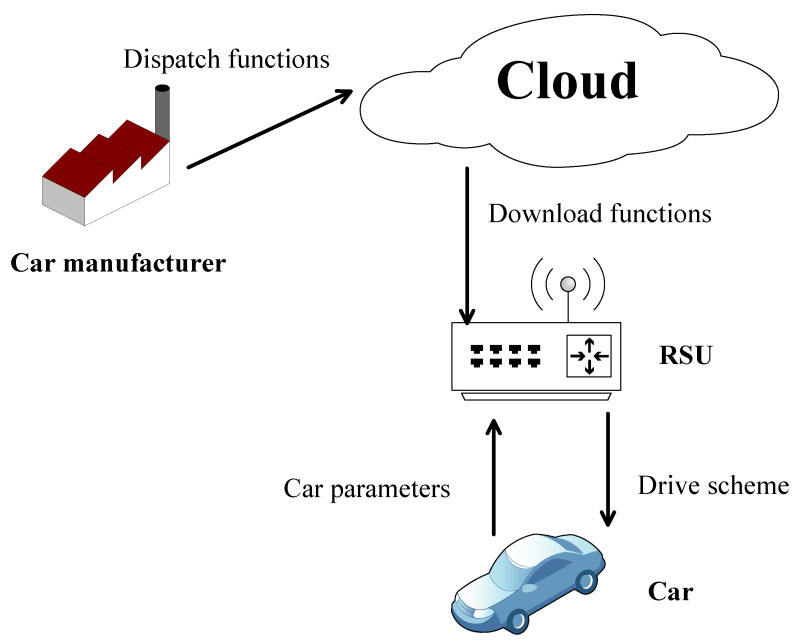
The architecture of a typical ITS application.

**Figure 2 sensors-22-04012-f002:**
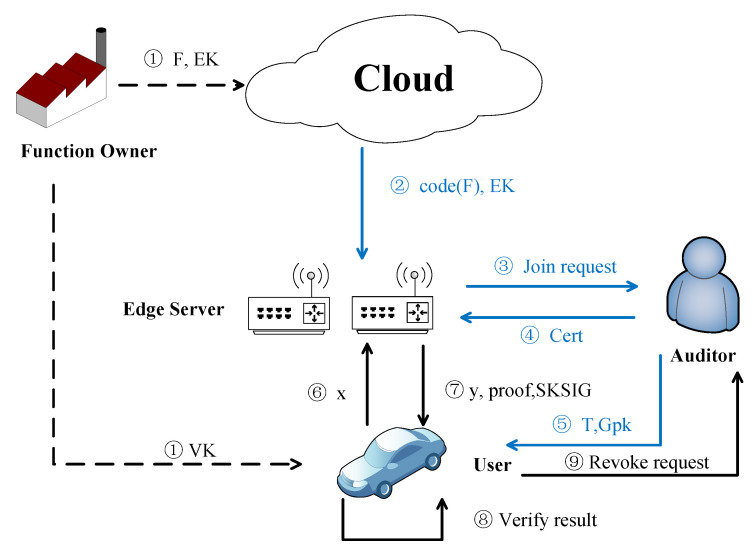
Model Architecture (Dotted lines stand for the initialize phase. Blue lines and black solid lines denote the join phase and outsource computing phase).

**Figure 3 sensors-22-04012-f003:**
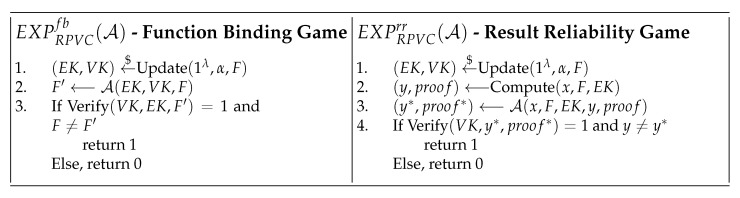
Security Games for RPVC.

**Figure 4 sensors-22-04012-f004:**
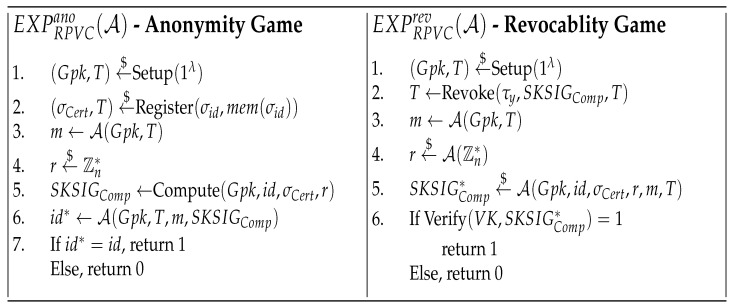
Security Games for RPVC.

**Figure 5 sensors-22-04012-f005:**
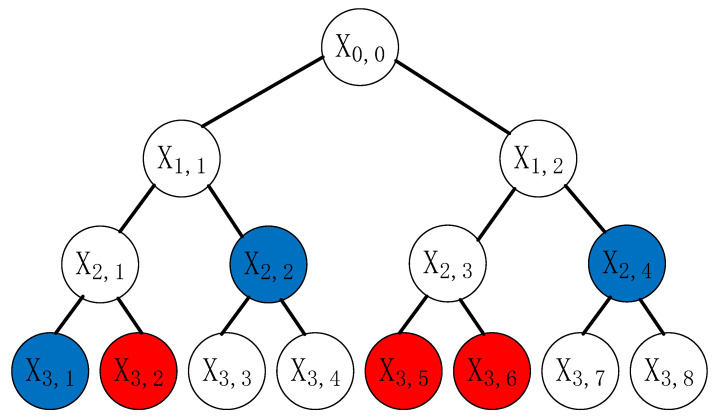
Example of SCST generation process. (The nodes to be revoked are represented in red, and the new subset covering the node is represented in blue).

**Figure 6 sensors-22-04012-f006:**
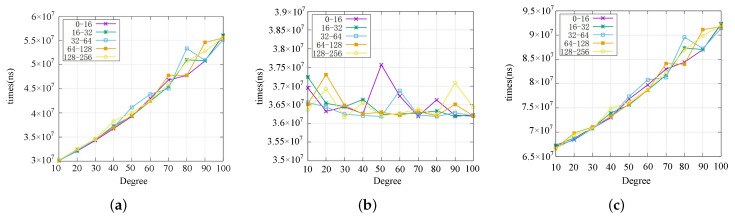
RPVC overhead. (**a**) Edge sever overhead. (**b**) User verify time. (**c**) Total delay.

**Figure 7 sensors-22-04012-f007:**
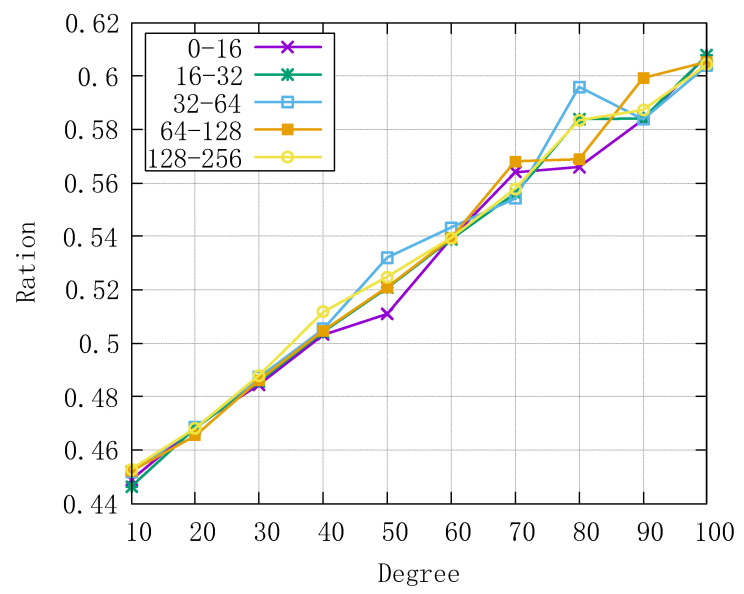
Proportion of edge server delay.

**Table 1 sensors-22-04012-t001:** Notations.

Symbol	Definition	Symbol	Definition
Gsk	Group private key	Gpk	Group public key
*T*	Quick access binary tree	*L*	Edge server list
$\sigma_{id}$	Mask code of user’s id	$mem(\sigma_{id})$	Mask code of group membership
λ *F*	Security parameterFunction	α	Private random number choosen by function owner
VK	Verify Key	EK	Computing Key
Cert	Certificate	$\sigma_{Cert}$	Mask code of Certificate
*y*	Computing result	$SK_{S}$	Signature private key set
$\tau_{y}$	The token decides whether accept *y*	proof	Verifiable proof of *y*
SKSIG	Signature of knowledge for joining the group or computing	*x*	The value which user outsource.

**Table 2 sensors-22-04012-t002:** Comparison of RPVC with existing schemes.

Scheme	Auditable	Audit Method	Update Frequency	Key Generator	Cryptographic Algorithm
[42]	No	Cannot	Dynamic	Single	Symmetric
[43]	No	Cannot	Dynamic	Multi-party	Symmetric
[44]	Yes	Iterate list	Dynamic	Multi-party	ECC
[45]	Yes	Direct	Never	Single	BBS
[46]	Yes	Iterate list	Dynamic	Single	ECC
RPVC	Yes	Query the tree	Timed	Single	Zero-knowledge signature

**Table 3 sensors-22-04012-t003:** Execution Time of Four Stages of Group Signature (ms).

Register	Signature	Verify	Open
27.996	28.162	29.001	287.466

**Table 4 sensors-22-04012-t004:** Add/Delete An Edge Server.

	Scale	$2^{11}$	$2^{12}$	$2^{13}$	$2^{14}$	$2^{15}$
Test Content						
Size of SCST(KB)		5.974	5.340	5.340	5.023	4.709
Execution time(ms)		0.085	0.158	0.270	0.524	0.986

**Table 5 sensors-22-04012-t005:** Cumulative SCST Size.

Edge Server Scale	$2^{11}$	$2^{12}$	$2^{13}$	$2^{14}$	$2^{15}$
Size(MB)	1.213	2.363	4.352	7.703	14.785

## Data Availability

The datasets generated for this study are available on request to the corresponding author.

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
