# Peer review of "RPVC: A Revocable Publicly Verifiable Computation Solution for Edge Computing"

_sensors, 2022, doi:10.3390/s22114012_

Round 1
Reviewer 1 Report
The manuscript presented by the authors touches on a current topic and the reviewer believes that the publication will be interesting to readers.
The reviewer has the following remarks:
- Number all formulas used;
- To present the sources of the used formulas;
- The authors to answer the question "Did their research have restrictive conditions?"
- The conclusions made should be supported by the results of the research.
- To present new research on the issues under consideration. Only a few of the presented references are from the last 5 years. The issues under consideration are topical and it is good to present the world research in the last few years more objectively.
- To specify and emphasize the practical applications of the research.
Author Response
Point 1: Number all formulas used.
Response 1: Thank you for your criticism and correction. I have numbered all formulas used, corresponding to comments 11~20 in RPVC(with comments).pdf.
Point 2: To present the sources of the used formulas.
Response 2: This problem is modified together with the previous one. I have presented the sources of all used formulas, corresponding to comments 11~20, 22, and 24 in RPVC(with comments).pdf.
Point 3: The authors to answer the question "Did their research have restrictive conditions?"
Response 3: Thank you for such a profound question, the main restrictive condition of our research is the outsourced computing functions which evaluated by edge servers can only be in polynomial form. Although some schemes can calculate any form of function, users in these schemes need to perform heavy verification calculations and store a large number of verification data locally. These drawbacks make such public verifiable schemes don’t suit edge computing scenarios. In essence, the function that operates in the computer uses the numerical analysis method to interpolate and approximate the continuous function through the discrete method. Therefore, we restrict the outsourced computing function to polynomial form, other forms of functions will be supported in future work.
Point 4: The conclusions made should be supported by the results of the research.
Response 4: The previous conclusion of the article is indeed too simple. So, I rewrote the section “Conclusion”, it first introduces what requirements of edge computing do our scheme solves, then, list the experimental results to enhance the convincingness of the conclusion. Next, specify the practical applications. Finally, it provides summarized future work.
Point 5: To present new research on the issues under consideration. Only a few of the presented references are from the last 5 years. The issues under consideration are topical and it is good to present the world research in the last few years more objectively.
Response 5: Appreciate your criticism and correction, I added some state-of-the-art research to present the world research more objectively corresponding to comments 1~8. For the comparison of the proposed model with existing ones, I made a table and analyzed it in comment 43.
Point 6: To specify and emphasize the practical applications of the research.
Response 6: Our article is mainly a theoretical model. The practical application is to improve the existing IoV applications and assist in the construction of intelligent transportation. The practical applications of the research are provided in sections “Discussion” and “Conclusion”, which correspond to comments 44 and 45 in RPVC(with comments).pdf.

Reviewer 2 Report
this paper is well written.
- The literature is not comprehensive. Please properly comment on the following papers:
- 1) Accelerating Edge Intelligence via Integrated Sensing and Communication
- 2) Distributed Dynamic Map Fusion via Federated Learning for Intelligent Networked Vehicles
- 3) Reconfigurable Intelligent Surface Assisted Mobile Edge Computing with Heterogeneous Learning Tasks
- The security proof is not rigorous. Please check it and write it in a logical way. Besides, I suggest providing intuitions between this security analysis. - The literature is not comprehensive. The typos should be corrected.
Author Response
Point 1: The literature is not comprehensive. Please properly comment on the following papers:
- 1) Accelerating Edge Intelligence via Integrated Sensing and Communication
- 2) Distributed Dynamic Map Fusion via Federated Learning for Intelligent Networked Vehicles
- 3) Reconfigurable Intelligent Surface Assisted Mobile Edge Computing with Heterogeneous Learning Tasks
Response 1: Thanks for your suggestion, I have read these three articles and find that each of them proposed excellent ideas, which makes me benefit a lot. I have cited them in References [50],[51], and [52]. Appreciate your recommended articles again.
Point 2: The security proof is not rigorous. Please check it and write it in a logical way. Besides, I suggest providing intuitions between this security analysis.
Response 2: The problem you pointed out is very critical. My previous security proof is hard to understand. Therefore, I fixed the error, adjust the format and add intuitions, as comments 23~28,30,32-41 in RPVC(with comments).pdf.
Point 3: The literature is not comprehensive. The typos should be corrected.
Response 3: Thank you for your criticism and correction. I did my best to revise the English in the full text. In addition, I added some state-of-the-art research corresponding to comments 1~8 in RPVC(with comments).pdf. For the comparison of the proposed model with existing ones, I made a table and analyzed it in comment 43.

Reviewer 3 Report
In the paper, authors proposed an RPVC model for the edge computing scenario. The RPVC model can not only ensure that the edge computing result is reliable, but also revoke dishonest edge servers. The author's proposed scheme reduces the publicly verifiable computational cost through polynomial promise and increases the efficiency of revocable group signature and execution through subset coverage. Their security case demonstrates that RPVC has features of functional binding, result reliability, anonymity, and revocability. Finally, experiments show that the cost of applying RPVC to existing IoV applications is acceptable.
I think the paper has its community and is relevant to it.
- There is a lack of state-of-the-art and a comparison of the proposed model with the existing ones in the literature.
- Future work with the proposed RPVC model needs to be included in the conclusion and also discussed in the paper in a section under "Discussion".
- The English language definitely needs to be worked on and improved.
Author Response
Point 1: There is a lack of state-of-the-art and a comparison of the proposed model with the existing ones in the literature.
Response 1: The problem you pointed out is very critical. My previous version can not objectively reflect the current research in the world. So, I added some state-of-the-art research corresponding to comments 1~8 in RPVC(with comments).pdf. For the comparison of the proposed model with existing ones, I made a table and analyzed it in comment 43.
Point 2: Future work with the proposed RPVC model needs to be included in the conclusion and also discussed in the paper in a section under "Discussion".
Response 2: Your suggestions do make the article more forward-looking. Therefore, I added a new section “Discussion” as comment 44 in RPVC(with comments).pdf. This section first summarizes the experimental results, then analyses the reason why our scheme can get the results, at last, the future work was given. In comment 45, section “Conclusion” also includes a summarized future work.
Point 3: The English language definitely needs to be worked on and improved.
Response 3: Thank you for your criticism and correction. I did my best to revise the full text.

Round 2
Reviewer 2 Report
Thank you for your revision.